# Delayed Effect of Dry-Land Strength Training Sessions on Swimming Performance

**DOI:** 10.3390/jfmk8030087

**Published:** 2023-06-22

**Authors:** Alexandros Tsoltos, Gavriil Arsoniadis, Charilaos Tsolakis, Panagiotis Koulouvaris, Theocharis Simeonidis, Alexandros Chatzigiannakis, Argyris Toubekis

**Affiliations:** 1Sports Excellence, 1st Orthopedics Department, School of Health Sciences, National and Kapodistrian University of Athens, 12462 Athens, Greece; 2Division of Aquatic Sports, School of Physical Education and Sports Science, National and Kapodistrian University of Athens, 17237 Dafne, Greece; 3Sports Performance Laboratory, School of Physical Education and Sports Science, National and Kapodistrian University of Athens, 17237 Dafne, Greece; 4Pesristeri Swimming Club, 12137 Peristeri, Greece

**Keywords:** strength training, swimming, performance evaluation, biomechanical variables

## Abstract

The purpose of the study was to examine the effects of dry-land strength endurance (SE) and maximum strength (MS) sessions on next-day swimming performance. Eight swimmers (age: 18.6 ± 2.9 years) performed evening training sessions (19:00–19:40), including: (i) SE (2 × 15 − 20 repetitions, 50% of 1-RM), (ii) MS (2 × 5 repetitions, 90% of 1-RM), (iii) control (CON: no dry-land training). All sessions were followed by a 90-min swimming training (20:00–21:30). Medicine ball throw and countermovement jump, free countermovement jump and squat jump were evaluated before and after the dry-land training session and 12 h later, before a 100-m front crawl sprint (next day at 8:30 a.m.). Performance time, RPE, blood lactate and biomechanical variables in 100-m sprint were no different between conditions (time, MS: 64.70 ± 7.35, SE: 63.81 ± 7.29, CON: 64.52 ± 7.71 s, *p* > 0.05). Jump height was not changed before and after dry-land and before the 100-m sprint in all conditions (*p* > 0.05). Medicine ball throw was lower in MS compared to CON before the 100-m sprint (MS: 4.44 ± 1.11, vs. CON: 4.66 ± 1.21 m, *p* < 0.05). Upper-body but not lower- body muscle function may be affected by MS training. However, performance in a 100-m test is not affected by dry-land training performed 12 h earlier.

## 1. Introduction

Dry-land strength endurance (SE) or maximum strength sessions (MS) may be performed a few minutes prior to swimming training [1]. These priming dry-land sessions may deteriorate swimmers’ performance or technique during the following swimming training session [2,3]. However, in a pre-competition setting, especially during the general preparation phase of training, the swimmers participate in a session involving concurrent resistance training and swimming some hours before a competition. Evidence indicates that a morning concurrent dry-land and swimming session may enhance an afternoon swimmer’s 100-m performance time [4]. Moreover, 50-m front crawl performance was improved 24 h after a priming dry-land strength endurance and power training [5]. 

As such, priming dry-land strength training, even with heavy loads (≥85% of 1-repetition maximum), seems to be beneficial for athletic performance up to 48 h as a delayed potentiation effect [6]. Similar findings have been reported following strength training with heavy loads contributing to power athletes’ performance improvements and increments in their lower-body power output the following day [7]. The abovementioned findings indicate that when 6 to 24 h of recovery has been completed following a specific resistance training session, this may be beneficial to performance [4,5,6,7]. The recovery period after a strength training session with heavy loads may present a biphasic recovery of performance (i.e., 11 h and 11 to 22 h) that may be attributed to structural muscle changes (excitation–contraction coupling) [8]. Previous research indicated a higher energy cost during a running economy test following resistance training session and decreased performance the following day [9]. 

A 12-h period of recovery may be practically applicable to swimmers following an afternoon and a morning session or competition. This information about the effect of dry-land strength training on the following day, however, remains unknown in swimming and needs further research. The aim of the current study is to examine the acute effect of priming dry-land strength endurance and maximum strength training sessions on swimming performance the following day. We hypothesized that dry-land strength training performed 12 h before a 100-m sprint test would negatively affect performance compared with control condition (no dry-land training). 

## 2. Materials and Methods

### 2.1. Experimental Approach to the Problem 

A one-group repeated-measures design was applied in the study, including three experimental conditions. Swimmers with a counterbalanced order performed a control session and two equal total load dry-land strength training sessions; strength endurance (SE) and maximum strength (MS) in an afternoon training and performance in a 100-m sprint test was evaluated 12 h later (the following morning). Countermovement jump (CMJ), free countermovement jump with arm swing (FCMJ), squat jump (SJ) and medicine ball throw (MBT) were used to evaluate muscle function in the experimental sessions. All the experimental procedures were completed in three weeks, during the swimmers’ general preparation period. 

### 2.2. Participants 

Eight competitive swimmers (5 males and 3 females) from the same swimming club volunteered to participate in the study. Swimmers’ anthropometric and performance characteristics are shown in Table 1. Swimmers did not take any dietary supplements of any kind or medication, and they were free from injury. All the participants were familiar and had at least two years of experience with dry-land strength training protocols. The local institutional review board approved the experimental procedures (approval number: 1111), which were in accordance with the Declaration of Helsinki. All the participants and their legal guardians signed an informed consent following a detailed explanation of the experimental procedures. 

### 2.3. Testing Procedures 

#### 2.3.1. Preliminary Testing and Familiarization Session

During the first session, swimmers’ anthropometric characteristics were evaluated. Body mass, body height and arm-span were measured. Body mass index and body fat percentage were calculated using the Jackson and Pollock equations [10]. In the second session, swimmers’ one-repetition maximum (1RM) in bench press, seated pull rowing, (swimmers were permitted to move their torso during the pull), and half squat (knee angle at 90°), were measured using standard procedures (ICC = 0.96, 0.98 and 0.98, respectively) [11]. In the third and fourth sessions, the swimmers were familiarized with the dry-land strength exercises that were performed in the experimental conditions. Familiarization included the three main exercises—bench press, seated pull rowing, and half squat—performing 2 to 3 sets of 15 to 20 repetitions, with a preferred external load in each exercise and 2 sets of 15 to 20 repetitions of sit ups and back extensions as secondary exercises. The aim of the familiarization session was to standardize swimmers’ technique in dry-land strength training sessions.

#### 2.3.2. Experimental Conditions 

Each swimmer completed in a random order: (i) SE (2 × 15 − 20 repetitions, 50% of 1-repetition maximum); (ii) MS (2 × 5 repetitions, 85% of 1-repetition maximum); (iii) no dry-land (control: CON) in a 40-min afternoon session (18:00–18:40 p.m.). SE, MS and CON sessions were followed by the same content 90-min in-water swimming training (19:00–20:30 p.m.). Upper-body muscle performance was evaluated using a 3-kg medicine ball throw. Lower-body performance was evaluated by a CMJ, a FCMJ and a SJ, before and after each dry-land training session and the respective time moments in the CON condition. All upper- and lower-body performance tests were repeated 12 h later, in the next morning session and before a 100-m front crawl sprint test at 8:30 a.m. Heart rate (HR), rating of perceived exertion (RPE) and blood lactate concentration (BL) were measured before the start and after the 100-m front crawl sprint test. During 100-m sprint test, split time measurement (50-m), arm-stoke rate (SR), and arm-stroke length (SL) were calculated. The experimental procedures of the study were conducted in an outdoor 50-m swimming pool with a constant water temperature of 27 °C. The experimental protocol is illustrated in Figure 1.

#### 2.3.3. Dry Land Strength Training 

Both dry-land strength training sessions, SE and MS, consisted of the same five exercises that have been previously included in dry-land sessions for competitive swimmers [2,12]. The SE and MS training sessions’ characteristics are shown in Table 2.

The training volume of SE and MS sessions was equalized by manipulating the number of sets, number of repetitions, load/intensity and movement tempo during repetition, as shown in Equation (1).
(1)Training volume=Sets ×Repetitions ×%1−RM ×MT 
where %1-RM (repetition maximum) is the training load/intensity and MT is the movement tempo during a repetition in bench press, seated pulley rowing or half squat. The training volume for SE and MS sessions in arbitrary units is shown in Table 2. 

#### 2.3.4. Low Intensity Swimming Interval Training

Swimmers performed a low-intensity swimming interval training 20 min after the completion of the SE and MS sessions. Specifically, swimming training consisted of a 1000-m standardized warm-up (600-m choice swimming, 4 × 50-m front crawl drills, 4 × 50-m front crawl kicks); 6 × 200-m individual medley with 30 s of rest between repetitions while focusing on technique; a 2000-m main set (5 × 400-m front crawl) with a low intensity corresponding to a HR of 140 to 150 b·min^−1^ and a 30 s rest between repetitions; and a 300-m recovery/cool down period. The volume of the swimming training session was 4500 m, and the overall duration was about 90 min. 

### 2.4. Dry-Land Performance Evaluations 

#### 2.4.1. Upper Limb Evaluation 

Swimmers’ upper-limb performance was evaluated through medicine ball seated push throw distance (MBT) in a seated position, as described by Dorie et al. [13]. Initially, the swimmers were familiarized with the MBT procedure of performing two MBT trials. Then, swimmers performed a two-minute warm-up using the medicine ball with different weights (2 to 5 kg). Two minutes after the warm-up, the swimmers performed three exercises from a seated position with 60 s of rest between each trial using a 3-kg medicine ball (Amila, Greece). Two experienced researchers measured the horizontal distance of displacement in ball throwing. The average MBT distance was included in the statistical analysis. 

#### 2.4.2. Lower Limb Evaluation

CMJ [14], FCMJ with arm swing and SJ [15] were measured on a portable device (Optojump Next, Bolzano, Italy). The Optojump photoelectric system consists of two parallel bars positioned approximately one meter apart at the floor level. The bars were connected to a computer with the appropriate software that measures the flight time of vertical jumps with an accuracy of 1/1000 (1 kHz). After a standardized warm-up consisting of five minutes of low-intensity running, one warm-up trial in each jump technique and dynamic stretching, the swimmers performed three exercises in CMJ, FCMJ, and SJ. A 30-s and 120-s rest was allowed between each exercise and each jump technique, respectively. The average height displacement from each jump technique was used in the statistical analysis.

#### 2.4.3. Swimming 100-m Sprint on the following Day 

At 8:30 a.m. of the morning following each SE, MS and CON session a 100-m front crawl sprint test was conducted after a 1000-m warm-up. Immediately after the completion of the 100-m test, HR was recorded (Polar Electro, Kempele, Finland) and RPE was indicated (0–10 points Borg scale) [16]. A fingertip blood sample was collected before the start and three minutes after 100-m sprint test to measure blood lactate concentration (Lactate scout, Germany) [17]. During the 100-m test, SR was calculated as the time to complete three stroke cycles, and SL was calculated by the ratio of swimming speed to SR in each 50-m split. In addition, an experienced timekeeper recorded swimmers’ performance times (Casio HS-80, Hubei, China). 

### 2.5. Statistical Analyses 

The normal distribution of the data was tested using the Kolmogorov Smirnov test. The sphericity was verified using the Mauchly test. When the assumption of Sphericity was not met, the significance of *F* ratios was adjusted according to the Greenhouse-Geiser procedure. Analysis of variance on repeated measures was used to compare BL, SR, SL, CMJ, FCMJ, SJ and MBT (three conditions × time points). A Tukey honest significant difference as a post-hoc test was used to compare the means when significant *F*-ratios were found. In addition, a one-way analysis of variance was used to compare the performance times in the 100-m front crawl test, HR and RPE. To estimate the size of the main effects and interaction, the partial eta squared (*η*^2^) values from the analysis of variance were used. Considering the sample size (N = 8), an effect size *d* of 0.80 was required to obtain a statistical power greater than 0.85 [18]. ICC using 1-way random effects was used to test reliability. Data are presented as mean and SD. Statistical significance was set at *p* < 0.05.

## 3. Results

### 3.1. Dry-Land Performance Evaluations 

The MBT distance was no different between conditions (*F*_2,14_ = 1.22, *η*^2^ = 0.15, *p* = 0.32) and between the time points of measurement (*F*_2,14_ = 2.67, *η*^2^ = 0.27, *p* = 0.10). However, the MBT distance was decreased by 4.4 ± 8.2% the following day, before the 100-m test, after MS compared to CON condition (*F*_2,14_ = 4.50, *η*^2^ = 0.39, *p* = 0.01, Figure 2). CMJ, FCMJ and SJ were higher in SE compared to MS, while no difference was found compared to CON (*p* < 0.05, see Figure 2). In addition, CMJ, FCMJ and SJ were similar at all time points of measurement (*p* > 0.05, Figure 2). 

### 3.2. Swimming Performance and Kinematics in 100-m Sprint Test 

The mean performance time in the 100-m front crawl test was similar between conditions (*F*_2,14_ = 0.58, *η*^2^ = 0.08, *p* = 0.57, Table 3). Similarly, there was no difference between conditions in each 50-m split time during the 100-m front crawl test (*F*_2,14_ = 0.58, *η*^2^ = 0.08, *p* = 0.57). The second 50-m split time was increased compared to the first in all conditions (SE: 8.6 ± 3.8, MS: 8.8 ± 4.2, CON: 7.0 ± 2.9 %, *p* < 0.05, Table 3). Mean SR and SL were similar between conditions (SR: *F*_2,14_ = 0.48, *η*^2^ = 0.07, *p* = 0.62; SL: *F*_2,14_ = 1.87, *η*^2^ = 0.21, *p* = 0.19). SR and SL were decreased in the second 50-m test compared to the first in all conditions (SR; SE: 4.0 ± 5.1, MS: 4.8 ± 4.4, CON: 4.6 ± 5.37%, *p* < 0.05, SL; SE: 4.6 ± 4.4, MS: 4.2 ± 3.3, CON: 2.9 ± 4.8, *p* < 0.05, Table 3). 

### 3.3. Physiological Variables and Rate of Perceived Exertion 

The baseline BL values that were measured 12 h after the last endurance swimming training prior to the 100-m time trial were similar in all conditions (*p* < 0.05, Table 4). Moreover, BL was similar between all conditions after the 100-m sprint test (*F*_2,14_ = 0.62, *η*^2^ = 0.08, *p* = 0.55, Table 4). Similarly, HR was no different between conditions (*F*_2,14_ = 1.95, *η*^2^ = 0.14, *p* = 0.33, Table 4). RPE obtained after the 100-m time trial was similar between the SE, MS and CON conditions (*p* > 0.05, Table 4). 

## 4. Discussion

The purpose of the study was to examine the effect of SE and MS training sessions, compared to CON, on performance in a 100-m sprint test on the following day (12 h later). Lower-limb performance, as indicated by the jump height in CMJ, FCMJ and SJ, was not affected by the previous day’s concurrent training. Moreover, upper-limb performance decreased in MS compared to CON; however, the 100-m performance time and physiological and biomechanical variables were similar between SE, MS and CON conditions. Swimmers may include a dry-land strength session of moderate load 12 h prior to a race. 

### 4.1. Dry-Land Performance Evaluations 

Considering the lower-limb evaluation, the jump height did not change before and after dry-land and before the 100-m test in all conditions. Findings from previous studies indicated decrements following MS training in CMJ (13%) in sedentary and active adults [19,20]. In another study, when a low volume power-type resistance training session was performed, a 3–5% increment was noticed in CMJ at both 24 and 48 h later [7]. It is possible that in our study any central or peripheral fatigue induced by the SE and MS sessions was not sustained 12 h later (the next morning) [21]. This has also been observed in a previous study where CMJ remained similar after MS or power-type strength training was performed 24 h later [22]. Possibly, the number of dry-land strength training exercises for lower limbs (1 exercise) performed in SE and MS sessions was not enough to affect some aspects of muscle function (i.e., maximal voluntary contraction) [21,22]. It is common practice for swimmers to perform dry-land exercises for their upper and lower body, similar to that presented in the study (Table 2). Subsequent to these exercises, it is expected that swimmers may increase their propulsive force and consequently improve their performance, especially in short and medium swimming distances (i.e., 50 m, 400 m) [23]. We may also assume that the recovery period (12 h), including a night’s sleep, was adequate for the restoration of all fatigue-inducing factors. Furthermore, the competitive level of swimmers as well as their training experience in this type of dry-land strength training exercises should be acknowledged to explain the unaffected jumping ability following SE or MS training. 

On the contrary, with regards to upper limbs, medicine ball throw distance decreased in MS compared to CON 12 h later, a finding that differs from the results of previous studies [24,25]. However, a higher volume of training was performed in the present study for the upper limbs (two exercises) compared to previous studies [25]. Possibly, this higher volume of training of the upper limbs forced swimmers to activate more type II muscle fibers during the MBT test. Thus, it is possible that MS led to a higher muscular perturbation (excitation–contraction coupling) and that a recovery period longer than 24 h was needed [8,22,26]. However, further studies with swimmers are required to understand the possible muscular changes following the completion of SE and MS sessions.

### 4.2. Swimming Performance on the following Day

Performance time in the 100-m sprint test the following day was similar between the SE, MS and CON conditions. A 2.5 ± 2.4% improvement in swimming performance in a 50-m front crawl sprint was observed 24 h after the completion of a dry-land strength endurance and power training [5]. Controversial findings were reported in runners when they performed a running economy test 12 h after the previous day’s concurrent training [9]. Authors attributed the runners’ higher energy cost and consequently performance impairment on the accumulation of fatigue the previous day [9]. In the present study, however, muscle function only partially deteriorated in MBT (upper limps) the day after MS condition, but this was not transferred to the 100-m test performance and kinematic variables. This finding indicates that any metabolic or neuromuscular disturbance of previous concurrent resistance and swimming training sessions returned to the baseline level during the 12-h recovery period. Perhaps the low-intensity swimming training that swimmers performed approximately 12 h prior to the 100-m time trial and following the completion of SE and MS training acted as an active recovery, facilitating the restoration of performance and kinematic variables [27]. It is also likely that the concurrent effect was more pronounced, masking any positive priming effects of the dry-land session. Another reason for the absence of difference between session may be the strength training protocol, which included a small number of strength training exercises and sets (three exercises) inducing a moderate load, which may not be enough to negatively affect swimming performance. A finding that is similar to a previous study of runners [28]. However, there are some limitations of the present study that should be mentioned. The swimmers that participated in the present study were not professional. The inclusion of both male and female, as well as the small sample size of swimmers that participated in the current study, should be acknowledged. Further, we did not manage to determine the swimmers’ sleep duration and quality of their sleep prior to the following day. 

## 5. Conclusions

Despite any muscular perturbation observed in the evaluation of the upper limbs 12 h following concurrent MS and swimming sessions, swimming performance in a 100-m sprint was not affected. It is likely that the content and characteristics of dry-land SE and MS sessions or the swimming training load of the previous day were not strong enough stimulus for any physiological or neuromuscular perturbation 12 h later. Swimmers may perform a SE or MS training session 12 h before a competition in which a single 100-m race is swum. 

## Figures and Tables

**Figure 1 jfmk-08-00087-f001:**
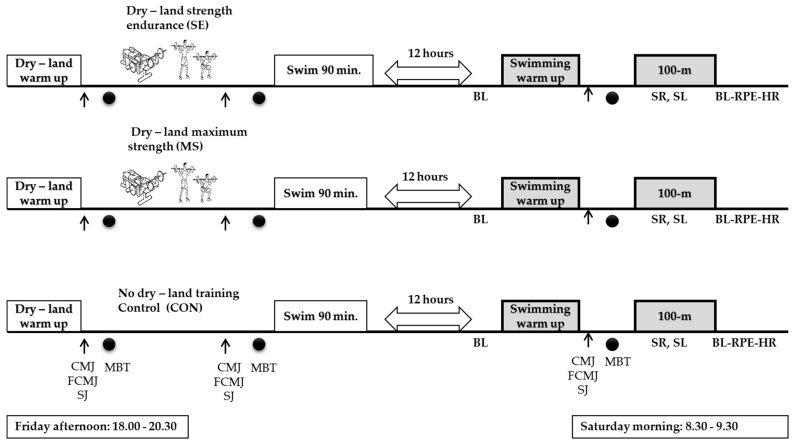
Experimental design of the study. ↑ indicates the testing evaluations of the lower limbs, such as countermovement jump (CMJ), free countermovement jumps (FCMJ) and squat jump (SJ), ● indicates the testing evaluation of the upper limbs, such as medicine ball throwing (MBT). BL: blood lactate concentration; SR: arm-stroke rate; SL: arm-stroke length; RPE: rate of perceived exertion; HR: heart rate.

**Figure 2 jfmk-08-00087-f002:**
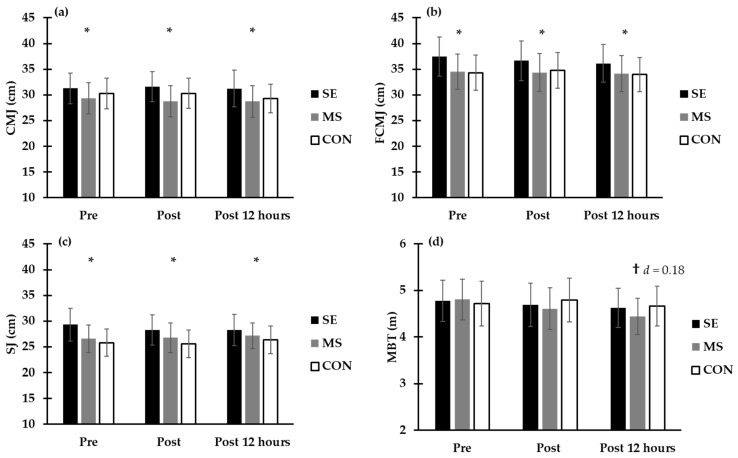
Acute effect of three experimental conditions, dry-land strength endurance (SE), dry-land maximum strength (MS) and control (CON) in dry-land performance evaluation of lower limbs in three different techniques: countermovement jump (CMJ; **panel a**); free countermovement jump (FCMJ; **panel b**); squat jump (SJ; **panel c**); and for upper limbs in medicine ball throw (MBT; **panel d**) in three time points of measurement: before dry-land strength training sessions (pre), after strength training (Friday afternoon; post) and 12 h later (Saturday morning; post 12 h). * *p* < 0.05, SE compared to MS in any time point, † *p* < 0.05, MS compared to CON.

**Table 1 jfmk-08-00087-t001:** Anthropometric, performance and training characteristics of swimmers who participated in the study.

Variables	Overall swimmers (n = 8)	Male Swimmers (n = 5)	Female Swimmers (n = 3)
Age (years)	18.6 ± 2.9	19.6 ± 3.1	17.7 ± 2.2
Body mass (kg)	65.6 ± 10.2	67.9 ± 5.6	64.6 ± 3.4
Body height (cm)	172.4 ± 6.4	174.6 ± 3.3	168.6 ± 2.8
Arm-span (cm)	176.2 ± 8.2	177.4 ± 4.5	171.6 ± 3.2
Seated height (cm)	90.4 ± 4.3	91.8 ± 3.6	90.5 ± 4.5
Body fat (%)	16.9 ± 4.0	15.5 ± 2.2	19.4 ± 2.1
Body mass index (kg/m^2^)	21.4 ± 2.3	22.0 ± 2.1	21.7 ± 1.9
100-m sprint time (s)	60.5 ± 7.7	56.3 ± 2.4	65.3 ± 3.2
FINA points (100-m front crawl)	555.6 ± 12.1	590.4 ± 14.5	497.6 ± 12.3
Competitive experience (years)	9.8 ± 1.6	10.2 ± 2.3	9.3 ± 1.5
Dry-land training experience (years)	2.0 ± 2.2	2.3 ± 1.6	2.0 ± 1.0

FINA: Fédération Internationale de Natation Amateur.

**Table 2 jfmk-08-00087-t002:** Detailed description of the dry-land strength endurance and maximum strength training sessions performed by the swimmers twelve hours prior to 100-m sprint test performed during the next day’s morning session. 1RM: one-repetition maximum.

	Dry-Land Strength Endurance Training Session (SE)				
Exercises	Number of Sets	Number of Repetitions	Intensity (%1-RM)	Rest	Movement Tempo/Repetition
Bench press	2	20	55	20 s	2 s/repetition
Seated pulley rowing	2	20	55	20 s	2 s/repetition
Sit-ups	3	15	Body weight	30 s	Preferred
Back extension	3	15	Body weight	30 s	Preferred
Half squat (knee angle 90°)	2	20	55	20 s	2 s/repetition
Overall duration	20 min				
	Dry-Land Maximum Strength Training Session (MS)				
Exercises	Number of Sets	Number of Repetitions	Intensity (%1-RM)	Rest	Movement Tempo/Repetition
Bench press	3	4	90	3 min	4 s/repetition
Seated pulley rowing	3	4	90	3 min	4 s/repetition
Sit-ups	3	15	Body weight	30 s	Preferred
Back extension	3	15	Body weight	30 s	Preferred
Half squat (knee angle 90°)	3	4	90	3 min	4 s/repetition
Overall duration	26 min				

2 s/repetition: 1 s lifting and 1 s lowering, 4 s/repetition: 2 s lifting and 2 s lowering.

**Table 3 jfmk-08-00087-t003:** Performance time and changes in biomechanical variables during the 100-m front crawl sprint test on the day following SE, MS and CON sessions.

Variables	SE	MS	CON
	Swimming Performance (s)		
Overall, 100-m	63.81 ± 7.29	64.70 ± 7.35	64.52 ± 7.71
1st 50-m split	30.48 ± 3.96	30.86 ± 3.84	30.97 ± 3.63
2nd 50-m split	33.33 ±3.40 *	33.84 ± 3.65 *	33.56 ± 4.13 *
	Arm-stroke rate (cycles·min^−1^)		
Overall, 100-m	42.60 ± 5.68	42.84 ± 5.59	43.33 ± 5.54
1st 50-m split	43.61 ± 6.51	43.89 ± 5.60	44.42 ± 6.11
2nd 50-m split	41.60 ± 5.02 *	41.79 ± 5.80 *	42.24 ± 5.27 *
	Arm-stroke length (m·cycle^−1^)		
Overall, 100-m	2.24 ± 0.13	2.20 ± 0.09	2.18 ± 0.13
1st 50-m split	2.30 ± 0.17	2.15 ± 0.09	2.22 ± 0.17
2nd 50-m split	2.19 ± 0.10 *	2.25 ± 0.11 *	2.15 ± 0.10 *

SE: dry-land strength endurance, MS: dry-land maximum strength, CON: control, *, *p* < 0.05: between 2nd to 1st split for each condition,

**Table 4 jfmk-08-00087-t004:** Blood lactate concentration, heart rate and rating of perceived exertion measured in 100-m front crawl sprint in three experimental conditions, following dry-land strength endurance (SE), dry-land maximum strength (MS) and control sessions (CON).

Conditions	Blood Lactate (mmol·L^−1^)	Heart Rate (b·min^−1^)	RPE (a.u)
SE	Before 100-m: 1.6 ± 0.9 After 100-m: 10.8 ± 4.3 †	178 ± 9	8.6 ± 1.5
MS	Before 100-m: 1.5 ± 0.2 After 100-m: 10.2±2.7 †	179 ± 10	9.0 ± 1.1
CON	Before 100-m: 1.3 ± 0.2 After 100-m: 9.7±3.9 †	175 ± 13	8.9 ± 1.0

† *p* < 0.05: compared to before 100-m.

## Data Availability

The data will be made available upon reasonable request to the corresponding author.

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
