# Peer review of "Delayed Effect of Dry-Land Strength Training Sessions on Swimming Performance"

_jfmk, 2023, doi:10.3390/jfmk8030087_

Round 1
Reviewer 1 Report
Notes in the annex

Author Response
Response to Reviewer 1
JFMK
Revised Manuscript ID: JFMK-2398940. Paper title: Delayed effect of dry-land strength training sessions on swimming performance
Opinion on the study
Comment 1
The aim of the study was to determine the effect of additional sessions of strength endurance and maximum strength training on dry land on next-day swimming performance. The authors concluded that additional strength and endurance training on dry land, applied 12 hours before the 100 m race, may have a negative impact on the results compared to competitors who did not train on dry land the day before.
Response 1
Thank you for the comment. In fact, both interventions applied in the current study did not affect swimmers’ performance on the following day. We have clearly mentioned that in our conclusion (lines 300-306).
Comment 2
The authors made anthropometric, performance and training characteristics of the swimmers participating in the study, however, these results do not refer to the subject of the work, because there was no relationship between the characteristics and the results of tests on land and in water.
Response 2
We have measured anthropometric characteristics of swimmers to express their general profile and we have reported the performance level. This information is not correlated with the dependent variables of the study. This was not the purpose of the study.
Comment 3
Small dependencies, or rather their lack, may result from too small an experimental factor, taking into account that the study involved athletes who regularly train a discipline, so strength training of a not very high intensity for this group of athletes could be irrelevant, which was confirmed by the results.
Response 3
We agree with your comment, however, the dry land sessions applied were adjusted according to swimmers’ level of ability (1RM). We considered this content of practical relevance since coaches will prefer to apply a moderate, but not high-load and long duration dry land session the day before a competition.
Comment 4
The value of this work would be to show what training and on what load the day before the start in the water would be optimal, which is too poor and which is too demanding. (short, mediumandlongdistances).
Response 4
In the present study we have tested two different types of strength training, namely strength endurance and maximum strength. We absolutely agree with the reviewer that testing the optimal load for each one of the above dry land training types combined with in water training would be interesting and should be examined in a future study.
Comment 5
The work has a correct layout, sufficient literature support, as well as correct statistical analysis. However, it does not provide an answer as to how the coach should behave with the players the day before the start and what factors should determine this behavior.
Response 5
Thank you for your comment. We have reported in lines 240-241 that coaches may include such types of dry land resistance training the day prior to a competition.
Reviewer 2 Report
Delayed effect of dry-land strength training sessions on swimming performance
This is an interesting study on how a dry-land strength priming followed by 90 minutes low volume swimming may affect 100m time performance, upper and lower body explosiveness as well as physiological factors. The research question is novel and Authors have done a well organised training study. However, there are some major and minor issues that need to be addressed before the manuscript will be considered ready for publication. Below, I present my point by point comments.
Major:
· Priming training sessions normally have duration approximately 30 minutes. Although strength training had duration 26 minutes, this priming session was followed by a 90-minutes swimming session which masked the positive results of dry-land priming. Then, why this design was performed? Is this design useful for preparatory phase only?
· Were there any differences between male and female swimmers? There are no comments for this inside the manuscript.
· It seems logical to me that when strength training is followed by a 90 minutes high volume swimming training session then the acute positive effects of strength training on neuromuscular performance will be lost. Although Authors clearly state in the manuscript that this is a standard training procedure during the preparation training phase, I think that this detail should be pointed out during the discussion as well. I would like to read Authors opinion about this since reading the discussion looks like Authors suggest that coaches may apply this training strategy during competitions as well.
· As a final major comment, why performance in 100m swimming remained unaltered? Lines 141-148 are solid hypothesis statements but were there any responders or no-responders swimmers (males and females)? Also, did the Authors searched about the influence of any physical ability attributes (strength, vertical jump, upper body power) on swimming performance? (https://doi.org/10.1007/s42978-023-00225-0 , Nishioka, T and Okada, J. Influence of strength level on performance enhancement using resistance priming. J Strength Cond Res 36(1): 37–46, 2022).
Abstract:
Abstract is well written and provides a good overview of the study. Just a small comment here concerning the training intensity of MS group. Authors present in the Abstract and inside the text that an 85% of 1-RM training intensity was applied. But in Table 2 a 90% of 1-RM is presented. Please, revise.
Introduction:
Intro is good, but not to analytical. There are strong review papers regarding priming on athletic performance by Harrison et al. (https://doi.org/10.1007/s40279-019-01136-3 , https://doi.org/10.1016/j.jsams.2019.09.010). I think Intro would be benefit from these studies. Also, there is a lack of research questions at the end of paragraphs. I suggest to Authors considering adding their research questions especially following each paragraph which will be related to the purpose of the study.
Also, there are no data in the into regarding the concurrent effect of resistance strength training with aerobic exercise. Maybe a small paragraph would help the hypothesis of the study concerning the expected negative benefits ον swimming performance.
Methods:
In the methods significant additions of details are needed.
In the experimental approach to the problem Authors should clearly state the experimental design of the study and whether experimental procedures were randomly assigned to participants as clearly presented in the experimental conditions paragraph. Also, how many days was the duration of the entire experimental procedures and how many days were given to swimmers between conditions. Was it a week?
Line 65: Authors present a Free CMJ. Is this CMJ with arms swing?
Table 1: Table1 is fine, although there seems to be an extra decimate digit in the SD value for body mass (overall swimmers) as well as for 100m performance (overall swimmers).
Lines 90-92: Familiarization included only three exercises and Table 2 presents 5 exercises. This would be confusing for readers. I suggest to Authors adding the abdominal and lower back exercises in the text in this point.
Line 97: Is it 85 or 90% of 1-RM (Table 2)?
Line 100: Here Authors report that upper body power performance was evaluated with a 3kg med-ball. However, in 2.4.1 paragraph a range of 2-5 kg med-balls was used. Please, clarify.
Lines 101-102: Abbreviations for vertical jumps have already been given.
Figure 1: Figure is good, but CMJs, SJ and MBT are missing from the image.
Question: Are RPE values refer to how the 90-minute swimming training session felt or for the overall training session including strength training and swimming?
Paragraph 2.3.3. Was there a guidance to participants concerning the velocity of movement during the concentric phase of movement (https://doi.org/10.1007/s40279-019-01239-x)?
Line 117: Here are five exercises.
Line 126: Was there a linear encoder to measure velocity?
Table 2. From my point of view the training intensity for SE group was too low. Maybe a 65-70% of 1-RM with 15 repetitions might have provided better results.
Page 6: Line 16: The mass of med-balls is different here. Also, during the test was there a randomized order of the balls thrown?
Line 21: Opto-jump is actually based more in the photocells technology than platforms. I suggest to Authors deleting the word platform.
Line 23: Change are to were.
Line 33: What was the range of RPE?
Line 26: Was there any warm-up vertical jump before maximum efforts?
Paragraph 2.4.2 : Why only the jumping height was calculated? Opto-jump provides power and Power per body mass as well.
Line 35: Authors need adding a reference here for the three minutes lactate blood sample.
Line 51: I haven’t seen any ICC index in the manuscript. I suggest to Authors adding these values after each measurement or maybe creating a Table.
Results:
A crucial question here is how SE induced significant greater increases in vertical jumps, a power oriented measurement compared to MS?
Line 73: There seems to be a split time measurement for 100m time-trial but Authors haven’t presented this in the methods.
Table 3: According to the text there is a significant difference for all the second 50-m split times compared to the first in all conditions. However, there is no index of significance in table 3.
Discussion:
Line 102: I suggest to Authors adding a small take home message at the end of the first paragraph.
Line 113: Delete may.
Line 114: Please, explain the statement “to impair the intramuscular environment”.
Paragraph 4.1: Reading the paragraph someone may assume that the training program was too easy for lower body and too hard for upper body. If this is the case, then what the Authors have to suggest to coaches and athletes to do in order to enhance neuromuscular activation, or avoid of doing to decrease neuromuscular activation.
Lines 141-144: Was the 90-minutes low intensity swimming session an active recovery or a high-volume training session masking the positive effects of dry-land on swimming performance? Was the concurrent effect more pronounced here?
Line 148: Before entering the conclusion paragraph, Authors should present the limitations of the study. Sample size, nutrition monitoring, sleep duration, mental mood, menstruation cycle for females, previous strength training experience etc.
Great work, well done.
Author Response
Response to Reviewer 2
JFMK
Revised Manuscript ID: JFMK-2398940. Paper title: Delayed effect of dry-land strength training sessions on swimming performance
This is an interesting study on how a dry-land strength priming followed by 90 minutes low volume swimming may affect 100m time performance, upper and lower body explosiveness as well as physiological factors. The research question is novel and Authors have done a well organized training study. However, there are some major and minor issues that need to be addressed before the manuscript will be considered ready for publication. Below, I present my point by point comments.
We thank you for the constructive comments that helped us to increase the quality of our manuscript. We have made corrections to comply with your suggestions. Below you will find a list of changes. Please note that the added or changed text is shown with track changes into the manuscript.
Major:
Comment 1
Priming training sessions normally have duration approximately 30 minutes. Although strength training had duration 26 minutes, this priming session was followed by a 90-minutes swimming session which masked the positive results of dry-land priming. Then, why this design was performed? Is this design useful for preparatory phase only?
Response 1:
Thank you for the comment. The swimming session that swimmers followed dry-land training sessions was conducted under low intensity endurance training. Then, we do not know if the positive results of dry-land session were masked. Besides, the aim of the study was to investigate the effects of dry-land training sessions in swimming performance and not the priming effect, so we believe that the design is useful for coaches annually.
Comment 2
Were there any differences between male and female swimmers? There are no comments for this inside the manuscript.
Response 2:
Thank you for the comment. The aim of the study was not to identify possible differences between genders. Likewise, the small sample size for female swimmers is a limitation to identify possible differences. Also, we have included this information as limitation in lines 293-298.
Comment 3
It seems logical to me that when strength training is followed by a 90 minutes high volume swimming training session then the acute positive effects of strength training on neuromuscular performance will be lost. Although Authors clearly state in the manuscript that this is a standard training procedure during the preparation training phase, I think that this detail should be pointed out during the discussion as well. I would like to read Authors opinion about this since reading the discussion looks like Authors suggest that coaches may apply this training strategy during competitions as well.
Response 3:
Thank you for the comment. We agree with your opinion that any priming effect will be lost 12 hours later especially when a long duration sleep is included. In the present study, the purpose was to examine the effect of concurrent SE or MS and swimming training on next day’s performance. We suggest that this approach with the present limitations may be applied by coaches, as it is not affected performance.
Comment 4
As a final major comment, why performance in 100m swimming remained unaltered? Lines 141-148 are solid hypothesis statements but were there any responders or no-responders swimmers (males and females)? Also, did the Authors searched about the influence of any physical ability attributes (strength, vertical jump, upper body power) on swimming performance? (https://doi.org/10.1007/s42978-023-00225-0 ,Nishioka, T and Okada, J. Influence of strength level on performance enhancement using resistance priming. J Strength Cond Res 36(1): 37–46, 2022).
Response 4:
Thank you for this interesting comment. We have examined the correlation of SJ, CMJ, FCMJ and MBT with performance difference in 100 m between MS-CON and SE-CON conditions. We found no correlation between these variables (r range from 0,00 to 0.20, p>0.05).
Abstract:
Comment 5
Abstract is well written and provides a good overview of the study. Just a small comment here concerning the training intensity of MS group. Authors present in the Abstract and inside the text that an 85% of 1-RM training intensity was applied. But in Table 2 a 90% of 1-RM is presented. Please, revise.
Response 5:
Thank you for the comment. This was corrected as you suggested (line 17).
Introduction:
Comment 6
Intro is good, but not to analytical. There are strong review papers regarding priming on athletic performance by Harrison et al. (https://doi.org/10.1007/s40279-019-01136-3 , https://doi.org/10.1016/j.jsams.2019.09.010). I think Intro would be benefit from these studies. Also, there is a lack of research questions at the end of paragraphs. I suggest to Authors considering adding their research questions especially following each paragraph which will be related to the purpose of the study.
Response 6:
Thank you for the comment. We have included papers regarding priming on athletic performance in lines (41-43) and (43-45). We also included hypothesis instead questions (lines 58-60) and the novelty of our study has already reported in lines 54-56.
Comment 7
Also, there are no data in the intro regarding the concurrent effect of resistance strength training with aerobic exercise. Maybe a small paragraph would help the hypothesis of the study concerning the expected negative benefits ον swimming performance.
Response 7:
Thank you for the comment. We have reported information about concurrent training effect in swimming (Arsoniadis [ref 2], Dalamitros [ref 3] see lines 32-34).
Methods:
In the methods significant additions of details are needed.
Comment 8
In the experimental approach to the problem Authors should clearly state the experimental design of the study and whether experimental procedures were randomly assigned to participants as clearly presented in the experimental conditions paragraph. Also, how many days was the duration of the entire experimental procedures and how many days were given to swimmers between conditions. Was it a week?
Response 8:
Thank you for the comment. That was corrected according to your suggestions (lines: 64 and 70-71).
Comment 9
Line 65: Authors present a Free CMJ. Is this CMJ with arms swing?
Response 9:
Thank you for the comment. We have included this information in line 68 paragraph “Experimental approach to the problem” and also, in the paragraph “Lower limps evaluation” (line 158).
Comment 10
Table 1: Table1 is fine, although there seems to be an extra decimate digit in the SD value for body mass (overall swimmers) as well as for 100m performance (overall swimmers).
Response 10:
Thank you for the comment. This was corrected as you suggested.
Comment 11
Lines 90-92: Familiarization included only three exercises and Table 2 presents 5 exercises. This would be confusing for readers. I suggest to Authors adding the abdominal and lower back exercises in the text in this point.
Response 11:
Thank you for the comment. That was corrected as you suggested (lines: 96-97).
Comment 12
Line 97: Is it 85 or 90% of 1-RM (Table 2)?
Response 12:
The correct value 90% of 1-RM appears in Table 2.
Comment 13
Line 100: Here Authors report that upper body power performance was evaluated with a 3kg med-ball. However, in 2.4.1 paragraph a range of 2-5 kg med-balls was used. Please, clarify.
Response 13:
We have rephrased to clarify that a 3 kg medicine ball was used for testing (see line 154). A range of different mass medicine balls were used only for warm up and familiarization session according to swimmers’ ability.
Comment 14
Lines 101-102: Abbreviations for vertical jumps have already been given.
Response 14:
This was corrected as you suggested.
Comment 15
Figure 1: Figure is good, but CMJs, SJ and MBT are missing from the image.
Response 15:
A revised version of Figure 1 was inserted including all tests applied.
Comment 16
Question: Are RPE values refer to how the 90-minute swimming training session felt or for the overall training session including strength training and swimming?
Response 16:
RPE was indicating only after the 100 m test session.
Comment 17
Paragraph 2.3.3. Was there a guidance to participants concerning the velocity of movement during the concentric phase of movement (https://doi.org/10.1007/s40279-019-01239-x)?
Response 17:
We have changed the movement velocity to movement tempo as a better terminology. The swimmers were guided by one of the experimenters to maintain the required tempo using a stopwatch. The movement tempo that swimmers followed in each dry-land strength training presenting as off note in Table 2 (line 137).
Comment 18
Line 117: Here are five exercises.
Response 18:
There are the three main exercises (bench press, half squat and row pull) and the secondary exercise (sit-ups and back extension).
Comment 19
Line 126: Was there a linear encoder to measure velocity?
Response 19:
Unfortunately, there was not a linear encoder; however, the swimmers were guided by one of the experimenters to maintain the required tempo using a stopwatch.
Comment 20
Table 2. From my point of view the training intensity for SE group was too low. Maybe a 65-70% of 1-RM with 15 repetitions might have provided better results.
Response 20:
Thank you for your comment. The training intensity for strength endurance group was dictated by previous studies using strength endurance protocols in swimming and water polo (Arsoniadis et al., Acute and Long-Term Effects of Concurrent Resistance and Swimming Training on Swimming Performance.doi:10.3390/sports10030029 ; Dalamitros et al., The acute effects of different resistance training loads on repeated sprint ability in water polo players. doi:https://doi.org/10.5114/hm.2021.103293).
Comment 21
Page 6: Line 16: The mass of med-balls is different here. Also, during the test was there a randomized order of the balls thrown?
Response 21:
As we have explained in comment #13, different mass of med-balls was used for familiarization and warm up only. A 3 kg medicine ball was used for testing (line 154).
Comment 22
Line 21: Opto-jump is actually based more in the photocells technology than platforms. I suggest to Authors deleting the word platform.
Response 22:
This was corrected as you suggested (line 158).
Comment 23
Line 23: Change are to were.
Response 23:
This was corrected as you suggested (line 161).
Comment 24
Line 33: What was the range of RPE?
Response 24:
RPE was 0-10 and this is clarified in line 172.
Comment 25
Line 26: Was there any warm-up vertical jump before maximum efforts?
Response 25:
We have missed this information that is now included in line 163.
Comment 26
Paragraph 2.4.2 : Why only the jumping height was calculated? Opto-jump provides power and Power per body mass as well.
Response 26:
Thank you for your comment. It is true that opto-jump provides information about power. For practical relevance we examined the jump height.
Comment 27
Line 35: Authors need adding a reference here for the three minutes lactate blood sample.
Response 27:
Thank you for your comment. This was added as you suggested (line 174, ref: 17).
Comment 28
Line 51: I haven’t seen any ICC index in the manuscript. I suggest to Authors adding these values after each measurement or maybe creating a Table.
Response 28
ICC is included in paragraph 2.3.1 (line 92).
Results:
Comment 29
A crucial question here is how SE induced significant greater increases in vertical jumps, a power oriented measurement compared to MS?
Response 29:
Thank you for your comment. In lower limb evaluation it has been shown that jumping performance is higher pre, post and 12h post exercise. Maybe it was not the effect of the dry-land session that induced these important increases. It is unknown if the SE session was the main reason for these results.
Comment 30
Line 73: There seems to be a split time measurement for 100m time-trial but Authors haven’t presented this in the methods.
Response 30:
Thank you for your comment. This information is included in lines 112-113.
Comment 31
Table 3: According to the text there is a significant difference for all the second 50-m split times compared to the first in all conditions. However, there is no index of significance in table 3.
Response: 31
Thank you for the comment. We included this information in Table 3 as you suggested.
Discussion:
Comment 32
Line 102: I suggest to Authors adding a small take home message at the end of the first paragraph.
Response 32:
Thank you for the comment; we have included a take home message as you suggested in lines 240-241.
Comment 33
Line 113: Delete may.
Response 33:
This was corrected as you suggested.
Comment 34
Line 114: Please, explain the statement “to impair the intramuscular environment”.
Response 34:
Thank you for the comment. We have presented better this information into text in lines 253-254.
Comment 35
Paragraph 4.1: Reading the paragraph someone may assume that the training program was too easy for lower body and too hard for upper body. If this is the case, then what the Authors have to suggest to coaches and athletes to do in order to enhance neuromuscular activation, or avoid of doing to decrease neuromuscular activation.
Response 35:
Thank you for your comment. We have included this information in lines 254-258.
Comment 36
Lines 141-144: Was the 90-minutes low intensity swimming session an active recovery or a high-volume training session masking the positive effects of dry-land on swimming performance? Was the concurrent effect more pronounced here?
Response 36:
Thank you for your comment. The swimming session that followed was a low intensity session as mentioned in the text but it is unknown if it acted as a recovery for swimmers or it had a concurrent effect on them. This was added in the text in lines 288-289.
Comment 37
Line 148: Before entering the conclusion paragraph, Authors should present the limitations of the study. Sample size, nutrition monitoring, sleep duration, mental mood, menstruation cycle for females, previous strength training experience etc.
Response 37:
Limitations have been added in lines 293-297.
Comment 38
Greatwork, welldone.
Response 38
Thank you for your comment.
Reviewer 3 Report
Is the work constructed in an accurate manner, even if it is not clear what the objective is, i.e. do you want to consider this practice as useful? It would not appear that way from the results.
The conclusions should be rewritten by better emphasizing what the outcomes are and how they can be useful in common training practice
An important limitation of the work is not only the small sample, but also the level of the participants, the fat percentages are quite high when compared to top level swimmers, this should be underlined within the limits of the work.
It needs a revision of some periods and phrases
Author Response
Response to Reviewer 3
JFMK
Revised Manuscript ID: JFMK-2398940. Paper title: Delayed effect of dry-land strength training sessions on swimming performance
We thank the reviewer for the constructive comments that helped us to increase the quality of our manuscript. We have made corrections to comply with your suggestions. Below you will find a list of changes. Please note that the added or changed text is shown with red color as track changes into the manuscript.
Comment 1
Is the work constructed in an accurate manner, even if it is not clear what the objective is, i.e. do you want to consider this practice as useful? It would not appear that way from the results.
Response 1:
We have examined the effect of SE and MS session on the following day 100-m performance time. Such a dry-land sessions is a common practice prior to swimming training but it is questionable some hours before a competition.
Comment 2
The conclusions should be rewritten by better emphasizing what the outcomes are and how they can be useful in common training practice.
Response 2:
Thank you for the suggestion. We have included a take home message in lines 240-241 which is according to your suggestion.
Comment 3
An important limitation of the work is not only the small sample, but also the level of the participants, the fat percentages are quite high when compared to top level swimmers, this should be underlined within the limits of the work.
Response 3:
We have recognized the small sample size and we have reported it as a limitation of the present study (lines 293-295). The level of the swimmers in our study is indicated in Table 1 by FINA point system. The participants may be characterized as national level swimmers according to the categorization as proposed by McKay, et al., Int. J. Sports Physiol. Perf. 2022, 17, 317–331.https://doi.org/10.1123/ijspp.2021‐0451. Concerning body fat or body mass index similar values have reported in previous publication including competitive swimmers:
(Arsoniadis et al., 2022, https://doi.org/10.1123/ijspp.2021-0516;
Zacca et al., 2019, https://doi.org/10.1080/02640414.2019.1572434;
Almeida et al., 2020, https://doi.org/10.1007/s00421-020-04348-y).
Comment 4
It needs a revision of some periods and phrases
Response 4:
We have made an effort improve grammar across the text.
Round 2
Reviewer 1 Report
The authors reliably and substantively addressed the previous comments. After the introduction of corrections and additions, the work gained more volue and, in my opinion, is suitable for publication.
Author Response
We thank you for the time that you spent to review our manuscript, and for your constructive comments that helped us to improve the quality of our manuscript.
Reviewer 2 Report
no comments.
Author Response
We thank you for the time that you spent to review our manuscript, and for their constructive comments that helped us to improve the quality of our manuscript.
Reviewer 3 Report
The authors have answered my questions, one point still remains, i.e., the bf is not to be considered as a high-level athlete, so this must be underlined as a limitation, the works that the authors cite are published but do not refer to athletes of high level, I personally deal with high-level swimmers and the bf in men does not exceed 12%
Better but still improvable
Author Response
Comment 1
The authors have answered my questions, one point still remains, i.e., the bf is not to be considered as a high-level athlete, so this must be underlined as a limitation, the works that the authors cite are published but do not refer to athletes of high level, I personally deal with high-level swimmers and the bf in men does not exceed 12%.
Response 1
We agree with the reviewer that a lower percentage of body fat characterizes high-level swimmers. We have included this as a limitation as you have suggested (line: 292). We refer that the participants were “competitive swimmers” (line 73), because they were participated in our national championship, and we have not inserted any comment on high-level swimmers. The categorization of swimmers lavel was based on McKay et al., 2022 https://doi.org/10.1123/ijspp.2021‐0451).
Comment 2
English language is better but still improvable
Response 2
We have tried to improve it throughout the text.